
# MUSICA MetOp/IASI {H₂O,δD} pair retrieval simulations for validating tropospheric moisture pathways in atmospheric models

Matthias Schneider[1], Christian Borger[1], Andreas Wiegele[1], Frank Hase[1], Omaira E. García[2], Eliezer Sepúlveda[2,3], and Martin Werner[4]

[1]Institute of Meteorology and Climate Research (IMK-ASF), Karlsruhe Institute of Technology, Karlsruhe, Germany
[2]Izaña Atmospheric Research Center, Agencia Estatal de Meteorología (AEMET), Santa Cruz de Tenerife, Spain
[3]Atmospheric Optics Group (GOA), University of Valladolid, Valladolid, Spain
[4]Alfred Wegener Institute (AWI), Bremerhaven, Germany

*Correspondence to:* M. Schneider
(matthias.schneider@kit.edu)

**Abstract.** The project MUSICA (MUlti-platform remote Sensing of Isotopologues for investigating the Cycle of Atmospheric water) has shown that the sensor IASI aboard the satellite MetOp can measure the free tropospheric {H₂O,δD} pair distribution twice per day on a quasi global scale. Such data are very promising for investigating tropospheric moisture pathways, however, the complex data characteristics compromises their usage in the context of model evaluation studies. Here we present a tool that allows for simulating MUSICA MetOp/IASI {H₂O,δD} pair remote sensing data for a given model atmosphere, thereby creating model data that have the remote sensing data characteristics assimilated. This model data can then be compared to the MUSICA data.

The retrieval simulation method is based on the physical principles of radiative transfer and we show that the uncertainty of the simulations is within the uncertainty of the MUSICA MetOp/IASI products, i.e. the retrieval simulations are reliable enough. We demonstrate the working principle of the simulator by applying it to ECHAM5-wiso model data. The exemplary model study clearly reveals the large potential of the MUSICA MetOp/IASI {H₂O,δD} data pairs for evaluating modeled moisture pathways. The tool is made freely available in form of MATLAB and Python routines and can be easily connected to any atmospheric water vapour isotopologue model.

## 1 Introduction

A major challenge for climate system modeling is the insufficient understanding of tropospheric moisture pathways and their coupling to atmospheric circulation (e.g. Stevens and Bony, 2013). Promising opportunities for addressing this challenge are provided by the stable isotopologues of tropospheric water vapour. The ratio between the different isotopologues gives complementary information on processes related to moisture uptake, exchange, clouds and atmospheric transportation upwind of the detected air mass (e.g. Dansgaard, 1964; Gat, 2000; Yoshimura et al., 2004).

Particularly informative is the distribution of pairs of the humidity concentration and the ratios between different isotopologues (like the distribution of {H₂O,δD} pairs, e.g. Galewsky et al., 2005; Noone, 2012; González et al., 2016). Here we use





the $\delta$-notation, where the HDO/$H_2O$ ratio is expressed as $\delta D = \frac{HDO/H_2O}{VSMOW} - 1$ (with the Vienna Standard Mean Ocean Water, VSMOW $= 3.1152 \times 10^{-4}$, Craig, 1961).

Recently, there has been large progress in measuring and modeling tropospheric water vapour isotopologues (Schneider et al., 2016; Galewsky et al., 2016) and there is great potential for validating the modeled moisture pathways by means of water

isotopologue model-measurement comparisons. Space-based remote sensing observations of {$H_2O$,$\delta D$} pairs obtained from MetOp/IASI measurements are particularly interesting because they offer quasi global coverage of {$H_2O$,$\delta D$} pairs for each morning and each evening and for many years (Schneider et al., 2016). However, the characteristic of such remote sensing pair data is complex and strongly varies with varying surface and atmospheric conditions. Only if the full complexity of the {$H_2O$,$\delta D$} pair remote sensing data product is correctly taken into account, respective model-measurement comparison studies

will be useful (see the discussion in Sect. 7 of Schneider et al., 2016).

Many aspects of the complex nature of the remote sensing data are captured by the averaging kernel matrix $\mathbf{A}$, which relates the atmospheric state detected by the remote sensing retrieval (state vector $\hat{\boldsymbol{x}}$) to the actual atmospheric state (state vector $\boldsymbol{x}$):

$$\hat{\boldsymbol{x}} = \mathbf{A}(\boldsymbol{x} - \boldsymbol{x_a}) + \boldsymbol{x_a} \tag{1}$$

whereby $\boldsymbol{x_a}$ is the a priori (also called background) atmospheric state. For a correct usage of the remote sensing {$H_2O$,$\delta D$}

pairs, the averaging kernels are indispensable. They are an important part of the remote sensing product and are thus generally provided together with the retrieved states.

For any comparison of remote sensing data with atmospheric model data, however, we would need the averaging kernels that correspond to the modeled atmospheric state and not for the atmospheric state observed by the satellite. The reason is that atmospheric humidity fields strongly vary on small spatial scales and on short time scales. We cannot expect that the

averaging kernel obtained for a remote sensing observation made at a certain location (often an area smaller than $100\,km^2$) and time period (often less than 1 second) is representative for the atmospheric state that is simulated by an atmospheric model for a larger area and without considering the very short time scale variations. The inconsistency to the observational data will be particularly important when comparing satellite observations to global models (often compromised in their horizontal resolution) and to so-called free-run model simulations, which are not nudged to meteorological analyses, but instead produce

their own dynamics solely driven by the model physics. Actually, if we are interested in long-term changes of the water cycle, we need to work with coupled ocean-atmosphere models (free-run simulations) and not with model simulation nudged to prescribed ocean temperatures.

In order to produce model {$H_2O$,$\delta D$} pair data that have the same characteristics as the remote sensing {$H_2O$,$\delta D$} pair data, we need a retrieval simulator that predicts the averaging kernel $\mathbf{A}$ for a remote sensing retrieval if made for the conditions as

given in the model. The time needed for calculating the simulations should be as short as possible (we need a very fast calculation) in order to test a variety of different model runs on different scales and with different parameters and parameterisations. The comparison of the {$H_2O$,$\delta D$} pair distribution consists then in statistical comparisons. Concretely, we will do statistical analyses of the relations between the modeled {$H_2O$,$\delta D$} pair distributions and the modeled atmospheric situation. The same




statistical analyses will be made with the observational data ({$H_2O,\delta D$} pair remote sensing data and observations that capture the atmospheric situations) and serve as the reference for evaluating the relations found for the different model runs.

A retrieval simulator for the TES HDO product has been proposed by Field et al. (2012) for use with the NASA GISS ModelE general circulation model (Schmidt et al., 2005; Field et al., 2014). They work with look-up tables containing typical

remote sensing data characteristics for some selected categorical parameters. They found that their method only works for a limited number of surface and atmospheric conditions (mainly for those encountered at low latitudes and over sea surface). Our idea is to determine the data characteristics from physical principles and thereby reasonably simulating the characteristics for any condition. Furthermore, we want to simulate the averaging kernels for the full {$H_2O,\delta D$} pair state and disseminate our simulation routine to the modeling community in a way that it can be coupled to the output of any isotopologue enabled

model.

In this paper, we present a physics-based method for simulating the MUSICA MetOp/IASI {$H_2O,\delta D$} pair averaging kernels that would result for MetOp/IASI observations made during known surface conditions (skin temperature and emissivity) and known vertical profiles of atmospheric temperature and water vapour. Section 2 briefly discusses the physics behind a retrieval simulator and Sect. 3 presents our simulation approach. In Sect. 4 the validity of the simulated {$H_2O,\delta D$} pair averaging

kernels is demonstrated by comparing the simulated kernels to kernels resulting from actual retrievals. Section 5 applies the simulator in the framework of exemplary model-measurement inter-comparison exercises, thereby revealing the large potential of the MUSICA MetOp/IASI {$H_2O,\delta D$} pairs for validating tropospheric moisture pathways in atmospheric models. Section 6 gives a summary and outlook.

## 2  Thermal nadir radiances and averaging kernels for {$H_2O,\delta D$} pairs

A remote sensor measures radiances that inform about the conditions where the radiation has been emitted and about the conditions during radiative transfer. In order to simulate remote sensing retrievals the respective radiative transfer has to be understood. In this section we give a brief overview on the principles of radiative transfer of thermal nadir radiances and outline its relation to the sensitivity of the remote sensing system.

### 2.1  Radiative transfer for thermal nadir geometry

The radiances $L$ at frequency $\nu$ measured by a thermal nadir sensor at the top of the atmosphere (TOA) can be approximated as follows:

$$L(\nu) =$$

$$E_{\mathrm{srf}}(\nu)B(T_{\mathrm{srf}},\nu)e^{-\int_{z'=z_{\mathrm{srf}}}^{\mathrm{TOA}}\frac{1}{\mu}\sigma(\nu)c(z')dz'} + \int_{z=z_{\mathrm{srf}}}^{\mathrm{TOA}}\frac{1}{\mu}\sigma(\nu)c(z)B\big(T_{\mathrm{atm}}(z),\nu\big)e^{-\int_{z'=z}^{\mathrm{TOA}}\frac{1}{\mu}\sigma(\nu)c(z')dz'}dz$$

$$= L1\big(E_{\mathrm{srf}}(\nu),T_{\mathrm{srf}},\nu,z_{\mathrm{srf}},\mu,\sigma(\nu),c(z)\big) + L2\big(z_{\mathrm{srf}},\mu,\sigma(\nu),c(z),T_{\mathrm{atm}}(z),\nu\big) \qquad (2)$$





Equation (2) does not consider neither the downwelling atmospheric emissions that are reflected at the surface nor scattering processes. Both are of secondary importance over the spectral range of the MUSICA retrieval (1190-1400 $\mathrm{cm}^{-1}$).

Using these approximations, there remain two contributions. The term $L1(\nu)$ describes the radiances that are emitted from the Earth's surface and subsequently modified in the atmosphere by absorption. The term $L2(\nu)$ simulates the radiances that are emitted from the atmosphere and than modified by absorption at higher altitudes.

The term $L1(\nu)$ mainly depends on surface temperature ($T_{\mathrm{srf}}$) and the total column amounts of the atmospheric absorbers ($\int c(z)dz$, whereby $z$ is the variable for altitude). $B$ is the function for blackbody radiation (Planck function), which depends on temperature ($T$) and frequency ($\nu$). The parameter $E_{\mathrm{srf}}$ stands for surface emissivity, $\mu$ for the ratio between slant and vertical path through the atmosphere (most satellite sensors have varying viewing angles through the atmosphere) and $\sigma$ is the cross section for interaction between the radiation and an atmospheric trace gas. The term $L2(\nu)$ mainly depends on the atmospheric temperature and trace gas concentration profiles ($T_{\mathrm{atm}}(z)$ and $c(z)$, respectively).

Particularly important for understanding the remote sensing system's sensitivity are the Jacobians ($\partial L/\partial c$), which document how the radiances change by changing the trace gas concentration at a certain altitude. The Jacobian for frequency $\nu$ and due to concentration changes at altitude $z$ can be calculated by:

$$
\begin{aligned}
\frac{\partial L(\nu)}{\partial c(z)} &= \\
&\frac{L\big(\nu, c(z) + \Delta c(z)\big) - L\big(\nu, c(z)\big)}{\Delta c(z)} \\
&= \frac{L1\big(\nu, c(z) + \Delta c(z)\big) - L1\big(\nu, c(z)\big)}{\Delta c(z)} + \frac{L2\big(\nu, c(z) + \Delta c(z)\big) - L2\big(\nu, c(z)\big)}{\Delta c(z)} \\
&= \frac{\partial L1(\nu)}{\partial c(z)} + \frac{\partial L2(\nu)}{\partial c(z)}
\end{aligned}
\tag{3}
$$

We can separately calculate the Jacobians for the two contribution terms ($L1$ and $L2$).

## 2.2 Averaging kernels

A retrieval algorithm works with discretised vertical profiles (atmospheric trace gas concentrations and temperatures at $n$ numbers of atmospheric levels) and discretised spectra ($m$ numbers of spectral bins). The atmospheric state vector $\boldsymbol{x}$ samples the trace gas concentration and temperature profiles (for atmospheric level number 1 to $n$) and the measurement vector $\boldsymbol{y}$ the radiances (for the spectral bin number 1 to $m$). The vector $\boldsymbol{y}$ has $m$ entries and the vector $\boldsymbol{x}$ has $\{t \times n\}$ entries ($n$ entries for each of the $t$ trace gas or temperature profiles). In discretised form the Jacobian is typically represented by a matrix called $\mathbf{K}$ with dimension ($m \times \{t \times n\}$), whose entries document how the radiances at spectral bin $i$ (ranging from 1 to $m$) change due to a change of the trace gas concentration or of the temperature at a certain atmospheric level (indexed by $j$, which ranges from 1 to $\{t \times n\}$:

$$
K_{i,j} = \frac{\partial y_i}{\partial x_j}
\tag{4}
$$

The averaging kernel ($\mathbf{A}$) can be calculated from the Jacobian ($\mathbf{K}$), the measurement noise covariance ($\mathbf{S}_\epsilon$, dimension $\{m \times m\}$), and the a priori covariance ($\mathbf{S}_\mathbf{a}$, dimension $\{(t \times n) \times (t \times n)\}$, whose inverse is used as constraint during the





inversion process). According to Rodgers (2000) it is:

$$\mathbf{A} = (\mathbf{K}^T \mathbf{S}_\epsilon{}^{-1} \mathbf{K} + \mathbf{S}_\mathbf{a}{}^{-1})^{-1} \mathbf{K}^T \mathbf{S}_\epsilon{}^{-1} \mathbf{K} \qquad (5)$$

### 2.3 The {$H_2O$,$\delta D$} pair product

Reliable water isotopologue ratio retrievals are possible by a logarithmic-scale retrieval of $H_2O$ and HDO together with a
constraint of $\ln[\mathrm{HDO}] - \ln[\mathrm{H_2O}]$ (Schneider et al., 2006; Worden et al., 2006). This approach means an optimal estimation
retrieval of $(\ln[\mathrm{H_2O}] + \ln[\mathrm{HDO}])/2$ and $\ln[\mathrm{HDO}] - \ln[\mathrm{H_2O}]$, which are good proxies for $H_2O$ and $\delta D$ (Schneider et al., 2012).

However, the retrievals are generally performed in the {$\ln[\mathrm{H_2O}]$,$\ln[\mathrm{HDO}]$} basis system and the averaging kernels are
generated for this basis system. In order to visualize the characteristics of the {$H_2O$,$\delta D$} pair product the {$\ln[\mathrm{H_2O}]$,$\ln[\mathrm{HDO}]$}
retrieval outputs have to be transformed to the {$H_2O$,$\delta D$} proxy basis system ({$(\ln[\mathrm{H_2O}] + \ln[\mathrm{HDO}])/2$,$\ln[\mathrm{HDO}] - \ln[\mathrm{H_2O}]$}).
Further details on the {$H_2O$,$\delta D$} proxy states and the transformation to the proxy basis system are given in previous MUSICA
publications (e.g. Schneider et al., 2012, 2016, and references therein).

A full averaging kernel matrix ($\mathbf{A}$) in the {$H_2O$,$\delta D$} proxy state consists of four blocks, each of which is a {$n \times n$} matrix,
whereby $n$ is the number of the vertical atmospheric grid points used for the retrieval:

$$\mathbf{A} = \begin{pmatrix} \mathbf{A_{11}} & \mathbf{A_{12}} \\ \mathbf{A_{21}} & \mathbf{A_{22}} \end{pmatrix} \qquad (6)$$

Figure 1 depicts the rows of the four blocks of $\mathbf{A}$ taking a MUSICA MetOp/IASI retrieval for the tropical ocean ($8.858°$N;
$163.112°$E) as example. Shown is $\mathbf{A}$ for the direct retrieval output (the Type 1 product, left graphic) and for the data product
that has been a posteriori processed in order to support {$H_2O$,$\delta D$} pair studies (the Type 2 product, right graphic). The reason
for the a posteriori processing will become evident from the discussion at the end of this subsection.

The two blocks along the diagonal describe the direct responses, i.e. they represent the averaging kernels for $H_2O$ and $\delta D$.
The outer diagonal blocks describe the cross responses, whereby in Fig. 1 the respective x-axis are scaled, thereby accounting
for the different magnitudes of the $H_2O$ and $\delta D$ variations. We have to consider that $\ln[\mathrm{H_2O}]$ variations are one order of
magnitude larger than $\delta D$ variations. This means that the entries in the $\mathbf{A_{12}}$ block must be 10 times larger than entries in the
$\mathbf{A_{11}}$ block in order to be of similar importance. Vice versa, entries in the $\mathbf{A_{21}}$ block can be one order of magnitude smaller
than entries in the $\mathbf{A_{22}}$ block and still have a similar importance.

For the Type 1 product we observe that $H_2O$ and $\delta D$ have very different sensitivities (compare the Type 1 kernel entries of
$\mathbf{A_{11}}$ and $\mathbf{A_{22}}$). This means that for the Type 1 the $H_2O$ and the $\delta D$ products represent different atmospheric altitude regions
and using these data in the context of {$H_2O$,$\delta D$} pair distribution plots can lead to defective interpretations (see discussion in
the context of Sect. 7 of Schneider et al., 2016).

With the help of the aposteriori processing we can overcome this problem. The Type 2 product are the aposteriori processed
data. For this data product $H_2O$ and $\delta D$ are representative of the same atmospheric regions (compare the Type 2 kernel entries
of $\mathbf{A_{11}}$ and $\mathbf{A_{22}}$). The $H_2O$ and $\delta D$ Type 2 data can be directly used for {$H_2O$,$\delta D$} pair distribution studies. Readers that are





interested in more details about the the aposteriori processing are recommended to study Schneider et al. (2012), Wiegele et al. (2014), Schneider et al. (2016) and Barthlott et al. (2016).

## 3   Retrieval simulations

For a given atmospheric state the averaging kernels can be calculated according to Eq. 5 by using the same atmospheric radiative
transfer code that is used during our retrieval process (PRFFWD-nadir, Schneider and Hase, 2011). However, such line-by-line calculations are rather time consuming, thereby strongly handicapping comprehensive model-measurement comparison studies. For a time effective simulation of the averaging kernels we have to speed up the radiative transport calculations.

### 3.1   Simplified radiative transfer calculations

There are many different atmospheric trace gases that interact with infrared radiation. For simplification we only consider
$H_2O$ and HDO. This is justified, because within the spectral window used by the MUSICA MetOp/IASI retrieval the spectral signatures of $H_2O$ and HDO are clearly dominating over the signatures from all the other trace gases.

The cross section $\sigma(\nu)$ describes the spectral dependency of the interaction between the trace gas and the radiation. It is spectrally fine structured. Line-by-line radiative transport models simulate these structures by using spectroscopic parameters (intensity, pressure broadening, temperature dependency of broadening, etc.) that are specific for each single line (whereby the
spectroscopic parameters are collected in spectroscopic databases, e.g. Rothman et al., 2013). We simplify the situation by performing calculations according to Eq. (2) for 76 different spectral bins. For the different spectral bins we assume different cross section values ($\sigma$) for $H_2O$ and HDO. For 57 spectral bins we assume interactions of radiation and $H_2O$ molecules, whereby we use 57 different cross section values ranging from $1 \times 10^{-31} \mathrm{m}^{-2}$ to $2.4 \times 10^{-23} \mathrm{m}^{-2}$. For the rest of the spectral bins (19 bins) we assume interaction of radiation and HDO molecules, whereby the 19 different cross section values lie between
$1 \times 10^{-31} \mathrm{m}^{-2}$ and $2.6 \times 10^{-26} \mathrm{m}^{-2}$. We use much more spectral bins with $H_2O$ signatures than with HDO signatures and $\sigma$ values for $H_2O$ that span a much wider range than the values for HDO. This is congruent to the spectral window used for our MUSICA retrieval (1190-1400 $\mathrm{cm}^{-1}$).

Further simplifications are that we calculate the Planck function ($B$ in Eq. 2) only for a single frequency ($\nu = 1250 \mathrm{cm}^{-1}$) and we assume that the surface emissivity ($E_{\mathrm{srf}}$ in Eq. 2) has no dependency on frequency.
Even with these simplifications there are still a lot of calculations necessary to get the Jacobian matrix K for given surface and atmospheric conditions. For each of the $n$ atmospheric levels we have to calculate the Jacobians for $H_2O$ for all the 57 spectral bins, where we have interaction between $H_2O$ and radiation. This makes $28 \times 57$ calculations (28 is the number of atmospheric levels that we consider for satellite ground pixels at sea surface). Similarly, we have $28 \times 19$ calculations for HDO. The derivatives with respect to atmospheric temperatures have to be calculated for all the 76 spectral bins, which makes $28 \times 76$
calculations. Finally we need to calculate the derivatives with respect to surface temperature for each of the 76 spectral bins. In total $(2 \times 28 + 1) \times 76 = 4332$ calculations have to be performed.





### 3.2 Simulation of Jacobians

We made the simplified calculations for all the cloud free MUSICA IASI (IASI-A and IASI-B) retrievals of 10 Aug. 2014. These are about 300000 individual observation, i.e. we calculated $4332 \times 300000 \approx 1300$ Mio. individual Jacobians $K_{i,j}$. In addition and in analogy to Eq. 3 we separate $K_{i,j}$ into the contribution terms $K1_{i,j}$ and $K2_{i,j}$ ($K_{i,j} = K1_{i,j} + K2_{i,j}$).

Figure 2 illustrates six Jacobians calculated for the 300000 individual observations of 10 Aug. 2014. They are for two different exemplary line strength (top and bottom panels show calculations for a weak and strong line, respectively) and for three different exemplary altitudes (from the left to the right: 0.5-0.8 km above surface, 5 km a.s.l. and 8 km a.s.l.). The color code informs about the value of $K = K1 + K2$ and the contribution term $K1$ is plotted along the x- and the contribution term $K2$ along the y-axes.

The term $K1$ is the contribution of the surface radiation that has been absorbed in the atmosphere and it is always negative (higher atmospheric $H_2O$ concentration means increased atmospheric absorption of the surface radiation). The term $K2$ represents the contribution of atmospheric emissions and subsequent atmospheric absorptions at higher altitudes and it can be negative and positive. We observe that it is mainly positive for the weak exemplary line and for the surface-near atmosphere and mainly negative for the strong exemplary line or higher altitudes.

For the weak line there is a certain anti-correlation between the contribution terms $K1$ and $K2$, meaning that the variation in $K = K1 + K2$ is smaller than the variation in each single contribution term. However, for higher altitudes or stronger lines these anti-correlation is much weaker. In general both contributions are important and have to be considered in order to get the correct Jacobian $K = K1 + K2$. For strong lines and high altitudes $K$ is generally negative. Positive values are mainly achieved for the weak line for the surface near atmosphere and to a less extent also for the strong line, but then mainly for the altitudes of 5 and 8 km.

The two lines with different line strength give complementary information about the atmospheric state. This is shown in Fig. 3, where $K$ values achieved for the weak and the strong lines are plotted against each other. The plot is made for Jacobians of the surface near atmosphere, 5 km a.s.l. and 8 km a.s.l.. Especially for the surface-near atmosphere and the middle troposphere, there is no clear correlation. There are many situations where the $K$ values for the strong line are high and for the weak lines low and there are also many situations where it is the other way round. This means that sometimes the weak

line and sometimes the strong line ensures the sensitivity of the remote sensing system. The complementarity of lines with different strengths is essential for achieving reasonable sensitivity for many different atmospheric situations and for different atmospheric altitudes. The latter becomes evident by looking on the example situation of the tropical ocean (whose averaging kernel is plotted in Fig. 1). The Jacobians calculated for this situation are indicated by the black circles filled by white colour:

while for altitudes at and below 5 km a.s.l. the weak line provides the information ($K$ calculated from the weak line is much larger than $K$ calculated from the strong line), for higher altitudes it is the strong line ($K$ for the strong line is clearly larger than $K$ obtained for the weak line).

     Figures 2 and 3 reveal the complexity of the Jacobians for thermal nadir radiation. For some situations the emission from the surface absorption in the atmosphere is decisive (term $K1$) and for other situations the emissions in the atmosphere (term





$K2$) are responsible for the dominant signals. Furthermore, there are atmospheric situations where the weak lines provide dominating signals and other situations where the strong lines are important. Due to this complexity, retrieval simulations can only work if the main characteristics of radiative transfer according to Eq. (2) are taken into account. This explains why simpler methods (for instance, those suggested by Field et al., 2012) can only be used for a limited number of surface and atmospheric

situations.

### 3.3   Simulation of averaging kernels

According to Eq. (5) and using the Jacobians as obtained from our simplified calculations together with the retrieval constraint matrix $\mathbf{S_a}^{-1}$ and the measurement noise error matrix $\mathbf{S}_\epsilon$ we can simulate the full averaging kernels $\mathbf{A}$ for the $\{\ln[H_2O], \ln[HDO]\}$ basis system. A subsequent transformation to the $\{H_2O, \delta D\}$ proxy basis system gives us the simulated $\{H_2O, \delta D\}$ pair averag-

ing kernel.

#### 3.3.1   Simulation routine for MATLAB and Python

The function that performs the simulations are freely available in the supplement of this manuscript. We provide this routine for MATLAB and Python platforms.

The routine needs as input the model data of surface altitude (in meter above sea level), surface emissivity (unitless) and sur-

face temperature (skin temperature in K). Further required inputs are the vertical profiles of altitude (in meter above sea level), pressure (in hPa), temperature (in K) and humidity mixing ratios (in $\ln[ppmv]$) as well as the number of levels on which the profiles are provided. In addition, the routine also asks for the satellite angle (in degree, which is needed for calculating $1/\mu$ in Eq. 2), whereby we recommend to use $25°$, which is the median angle we found for the quality filtered MUSICA MetOp/IASI $\{H_2O, \delta D\}$ pair observations. The constraint matrix $\mathbf{S_a}^{-1}$ and the measurement noise error matrix $\mathbf{S}_\epsilon$ are automatically calcu-

lated.

The output of the routine is the $\{H_2O, \delta D\}$ proxy pair averaging kernel for the MUSICA MetOp/IASI Type 2 product, as explained in the context of Eq. (6), i.e. it is the kernel in the $\{(\ln[H_2O] + \ln[HDO])/2, \ln[HDO] - \ln[H_2O]\}$ basis system.

#### 3.3.2   Example of a simulated $\{H_2O, \delta D\}$ pair averaging kernel

Figure 4 depicts an example of an output of the routine. It shows the averaging kernel simulation for the situation that cor-

responds to the actual averaging kernel as shown in Fig. 1. Both Figures serve as an example for the agreement between the actual and the simulated averaging kernels.

We can measure $H_2O$ profiles (see kernel blocks $\mathbf{A_{11}}$ of the Type 1 product). Both actual and simulated kernels indicate sensitivity between $500\,m$ above the surface and about $13\,km$ a.s.l.. There are some minor differences close to the surface (where the simulated kernels indicate larger sensitivity than the actual kernels) and above $12\,km$ a.s.l. (where the actual kernels

indicate larger sensitivity than the simulated kernels).





The Type 2 kernels show the situation after applying the aposteriori correction to the actual and the simulated kernels, respectively. The a posteriori correction assures that $H_2O$ and $\delta D$ kernels are almost identical thus allowing {$H_2O$,$\delta D$} pair studies (kernel blocks $\mathbf{A_{11}}$ and kernel blocks $\mathbf{A_{22}}$ of the Type 2 product are almost identical). Actual and simulated kernels indicate peak sensitivities between 5 km and 9 km a.s.l., whereby the kernels are rather broad. A minor difference is that the

actual kernel indicates a peak sensitivity at 5.7 km a.s.l., whereas the simulated kernel indicates a peak sensitivity at 8 km. A further difference are the cross kernels $\mathbf{A_{21}}$, which are simulated too large. However, these cross kernels entries change from positive to negative values with 2-3 km and have thus have no impact on atmospheric signals that take place over broad layers.

## 4 Validation of the retrieval simulations

The reasonable agreement between actual and simulated kernels for the exemplary tropical ocean situation is encouraging

(compare Figs. 1 and 4), however, a global validation of the simulations is needed. This is provided in this section. We compare parameters obtained from the actual averaging kernels with parameters obtained from the simulated averaging kernels. This is done for all situations where the measured and the simulated spectra have a reasonable agreement. Therefore, we require that the residual (root mean square of the difference between measured and simulated spectra) is not larger than 6.5‰ of the maximum intensity in the fitted spectral region. Larger residuals are in disagreement with the assumption of a normal

distribution of the residual values and correspond very likely to cloudy conditions (that are not well recognized by the cloud filtering) or to incorrect assumptions of surface emissivity (for instance over desert areas). This fit quality filter has also been applied in Schneider et al. (2016). After this filtering there are still 275000 observations that can be used for validating the averaging kernel simulations. This large amount of observations cover all geophysical situations around the globe and allow for very robust validation studies.

### 4.1 Degree of freedom for signal (DOFS)

The DOFS (Degree Of Freedom of Signal) value is a measure for the information content in a remote sensing product. The higher the DOFS value the more independent is the product from the a priori assumptions. The DOFS value is calculated as the trace of the averaging kernel matrix. We evaluate the trace of the matrix block $\mathbf{A_{11}}$ of the Type 1 product, which represents the DOFS for the Type 1 $H_2O$ profiles. Concerning the Type 2 product we evaluate the trace of the matrix blocks $\mathbf{A_{11}}$ or $\mathbf{A_{22}}$,

both are almost identical and represent the DOFS for the Type 2 {$H_2O$,$\delta D$} pairs.

The correlations between the DOFS values obtained from the actual and the simulated averaging kernels are depicted in Fig. 5. The correlations are in good agreement with the 1-to-1 diagonal (black line) and the linear least squares fits produce regression lines with a slope that is very close to unity (red line). The coefficients $R^2$ are about 70% and 80% for the Type 1 and Type 2 DOFS, respectively.

Our simulations well capture the actual sensitivity of the remote sensing system. The dependency of the sensitivity on the surface and atmospheric conditions seems to be well-understood. The simulation allow the separation of conditions leading to low sensitivity from conditions leading to high sensitivity.





### 4.2 Averaging kernel effects

We focus on the Type 2 averaging kernels, because this is the product type of choice for {$H_2O,\delta D$} pairs analyses. The DOFS for the MUSICA MetOp/IASI Type 2 product is typically between 0.5 and 1.2, meaning that the data represent a broad layer (see also the Type 2 averaging kernels as depicted in Figs. 1 and 4). We cannot measure the vertical structure of {$H_2O,\delta D$} pair
distribution, but we can measure the {$H_2O,\delta D$} pair distribution of layers with a thickness of about 5 km. So we are interested in validating the sensitivity of the remote sensing with respect to {$H_2O,\delta D$} pair variations that take place over such broad layers. Since the $H_2O$ and $\delta D$ kernels of the Type 2 product are almost identical it is sufficient to estimate the averaging kernel effect for $H_2O$ or $\delta D$. Here we do it for $\delta D$.

     Variations over broad layers can be captured by a covariance matrix $\mathbf{S_{cov}}$. For calculating $\mathbf{S_{cov}}$ for $\delta D$ we assume $100^2\%o^2$
along the diagonal and then calculate the outer diagonal elements by assuming a a correlation length of 5 km. Furthermore, we decouple the boundary layer (the first 500-800 m above the surface) from the atmosphere above by assuming a correlation length of only 500 m between the boundary layer and higher altitudes. Figure 6 gives a visualisation of this covariance matrix $\mathbf{S_{cov}}$ for $\delta D$.

### 4.2.1 Sensitivity error

We are interested in the errors caused by the limited sensitivity of the remote sensing system for observing the broad vertical structures described by the covariance matrix $\mathbf{S_{cov}}$. These errors can be estimated by the error covariance matrix $\mathbf{S_{err}}$:

$$\mathbf{S_{err}} = (\mathbf{A} - \mathbf{I})\mathbf{S_{cov}}(\mathbf{A} - \mathbf{I})^T \qquad (7)$$

whereby $\mathbf{A}$ is the averaging kernel matrix (here the block $\mathbf{A_{22}}$ of the type 2 product) and $\mathbf{I}$ the identity matrix.

     In the following we focus on analysing the square root values of the diagonal elements of $\mathbf{S_{err}}$ for the three altitudes: 1500-
2000 m above surface (the lower troposphere), 5 km a.s.l. and 8 km a.s.l.. The square root values of these elements will be referenced to as $s_{err}(LT)$, $s_{err}(5\,km)$ and $s_{err}(8\,km)$, respectively.

     Figure 7 compares $s_{err}$ values and products obtained from calculations with the actual and the simulated averaging kernels. In total there are six panels. The right panel in each row of panels from the top to the bottom shows the comparison of $s_{err}(LT)$, $s_{err}(5\,km)$ and $s_{err}(8\,km)$ values. The black line indicate the 1-to-1 diagonals and the red line the regression lines obtained
from linear least squares fits.

     The simulated kernels correctly identify the altitudes around 5 km a.s.l. as the tropospheric region where $s_{err}$ is typically lowest. These are the altitudes where the remote sensing system has its best sensitivity. The $s_{err}(5\,km)$ values are below 50‰ for about 80% of all observations, whereas for $s_{err}(LT)$ and $s_{err}(8\,km)$ this is only the case for about 8% and 35% of all observations, respectively. For the three altitudes the $s_{err}$ values obtained from the actual and simulated kernels correlate well
along the 1-to-1 diagonal. The correlation coefficient $R^2$ are between 66% and 93%, meaning that the simulations well capture the variability in the remote sensing system's sensitivity. The slopes of the linear regression lines are close to unity.

     The remaining three panels (left panels in the middle and bottom row of panels of Fig. 7) compare differences between $s_{err}(LT)$, $s_{err}(5\,km)$ and $s_{err}(8\,km)$ values. For instance, the leftmost panel in the bottom row of panels shows $s_{err}(8\,km) -$





$s_{\mathrm{err}}(\mathrm{LT})$. This difference is positive when the sensitivity error is larger for the atmosphere at 8 km than for the lower troposphere (LT). Similarly, it is negative when the sensitivity error is larger for the LT than for 8 km. The value of $s_{\mathrm{err}}(8\,\mathrm{km}) - s_{\mathrm{err}}(\mathrm{LT})$ gives details on the altitude region with best sensitivity. Best sensitivity is achieved around the altitude of 5 km. A positive $s_{\mathrm{err}}(8\,\mathrm{km}) - s_{\mathrm{err}}(\mathrm{LT})$ indicates that altitudes below 5 km make also an important contributions. Vice versa a negative

$s_{\mathrm{err}}(8\,\mathrm{km}) - s_{\mathrm{err}}(\mathrm{LT})$ suggests increased importance of the altitudes above 5 km.

We observe a good correlation between the $s_{\mathrm{err}}(8\,\mathrm{km}) - s_{\mathrm{err}}(\mathrm{LT})$ values obtained from the actual and simulated kernels. The correlation is along the 1-to-1 diagonal, with a $R^2$ value of 80% and a linear regression line with a slope being close to unity. This good agreement demonstrates that the variations in the vertical structure of the averaging kernels are well captured by our simulations.

The leftmost panel in the middle row of panels and the central panel in the bottom row of panels compare differences of $s_{\mathrm{err}}(5\,\mathrm{km}) - s_{\mathrm{err}}(\mathrm{LT})$ and $s_{\mathrm{err}}(8\,\mathrm{km}) - s_{\mathrm{err}}(5\,\mathrm{km})$ and confirm the capability of our simulations of capturing detailed vertical structures of the averaging kernels.

### 4.2.2   Simulation error

The error we make by using the simulated kernel instead of the actual kernel for the interpretation of broad vertical structures

can be estimated as:

$$\mathbf{S_{sim}} = (\mathbf{A_{act}} - \mathbf{A_{sim}})\mathbf{S_{cov}}(\mathbf{A_{act}} - \mathbf{A_{sim}})^T \tag{8}$$

whereby $\mathbf{A_{act}}$ and $\mathbf{A_{sim}}$ are the actual and simulated averaging kernel matrix.

Figure 8 shows a statistical analysis of the square roots of the diagonal elements of $\mathbf{S_{sim}}$ that correspond to the altitudes LT, 5 km a.s.l. and 8 km a.s.l. (the respective square root values are in the following referred to as $s_{\mathrm{sim}}(\mathrm{LT})$, $s_{\mathrm{sim}}(5\,\mathrm{km})$ and

$s_{\mathrm{sim}}(8\,\mathrm{km})$). The panels show cumulative occurrences of the different $s_{\mathrm{sim}}$ values. The grey dots show the occurrences for all data (all about 275000 situations including those with $s_{\mathrm{err}}$ values above 50‰). For all altitudes and in 97.5% of all the situations $s_{\mathrm{sim}}$ is smaller than 20‰, meaning that the simulator is very precise and there are only very few cases where it fails, i.e. where $s_{\mathrm{sim}}$ is larger than 20‰, which is the typical uncertainty of the MUSICA MetOp/IASI $\delta$D remote sensing data for the detection of broad structures (Wiegele et al., 2014, please note that the cross dependency on $H_2O$ can be neglected when

broad layers are considered)).

The black dots show the cumulative occurrences for those situations where the remote sensing system is actually sensitive for atmospheric $\{H_2O,\delta D\}$ pairs, i.e. where $s_{\mathrm{err}}$ is smaller than 50‰. These are the actually interesting situations, because only for these situations the data are significantly different from the apriori assumption and can make a contribution. For these interesting situations the occurrence of significant simulation errors ($s_{\mathrm{sim}}$ above 20-30‰) is further reduced. Actually there is

no single $s_{\mathrm{sim}}(5\,\mathrm{km})$ value above 30‰.

In this context we would like to note that for the altitude of 5 km a.s.l. the sensitivity criterion of $s_{\mathrm{err}} < 50‰$ is fulfilled for most situations (for about 80% of all situations), whereas for the LT and for 8 km altitude it is only fulfilled for about 8%



and 35% of all situations, respectively (see also discussion in the context of Fig. 7). The rest of the manuscript focuses on the altitude of 5 km a.s.l..

### 4.2.3 Geographical distribution of the errors

In this section we examine the geographical patterns of $s_{\mathrm{err}}(5\,\mathrm{km})$ and $s_{\mathrm{sim}}(5\,\mathrm{km})$.

Figure 9 depicts the global distribution of $s_{\mathrm{err}}(5\,\mathrm{km})$ as obtained from the actual kernels (almost identical patterns are obtained for $s_{\mathrm{err}}(5\,\mathrm{km})$ if calculated from the simulated kernels, not shown). The left panel shows $s_{\mathrm{err}}(5\,\mathrm{km})$ obtained for morning observations (at about 9:30 local time) and the right panel for evening observations (at about 21:30 local time). There are no significant differences between the morning and the evening patterns.

     The $s_{\mathrm{err}}(5\,\mathrm{km})$ values are generally lower in the tropics than in middle and high latitudes. The largest $s_{\mathrm{err}}(5\,\mathrm{km})$ values

are found for the middle and high latitudes of the southern hemisphere, which is the winter hemisphere in August. This latitudinal dependency describes the average situation, however, it is important to note that there are also tropical observations with relatively large $s_{\mathrm{err}}(5\,\mathrm{km})$ values (e.g. in the tropical Atlantic east of Cuba $s_{\mathrm{err}}(5\,\mathrm{km})$ is between 30 and 50‰). Vice versa there are also middle latitudinal observations in the wintertime hemisphere with low $s_{\mathrm{err}}(5\,\mathrm{km})$ values (e.g. over South America between Argentina and Brazil we get $s_{\mathrm{err}}(5\,\mathrm{km})$ values below 20‰).

Figure 10 shows the global distribution of the simulation error at 5 km a.s.l. ($s_{\mathrm{sim}}(5\,\mathrm{km})$) for the situation where there is a significant sensitivity with respect to atmospheric $\{H_2O,\delta D\}$ pairs (i.e. the situations where the sensitivity criterion $s_{\mathrm{err}} < 50‰$ is fulfilled). In agreement to Fig. 8 we observe that $s_{\mathrm{sim}}(5\,\mathrm{km})$ is mostly below 20‰, which is much lower than the respective $s_{\mathrm{err}}(5\,\mathrm{km})$ values (please note that the colour-scale in Fig. 10 is the same as in Fig. 9). While $s_{\mathrm{err}}(5\,\mathrm{km})$ is very variable, the $s_{\mathrm{sim}}(5\,\mathrm{km})$ values have a very homogeneous distribution. This means that the differences in the sensitivity between

low and high latitudes (but also respective differences within the tropics or outer-tropics) that manifest in the inhomogeneous $s_{\mathrm{err}}(5\,\mathrm{km})$ fields are well captured by the simulations. The simulated averaging kernels well capture the different sensitivities and we can use the simulations for any observation around the globe.

     Largest ($s_{\mathrm{sim}}(5\,\mathrm{km})$) values are found for the evening observations over the continents. For instance, there are some situations where the evening value of $s_{\mathrm{sim}}(5\,\mathrm{km})$ over South Africa comes close to 30‰. However, there is no single situation for

which $s_{\mathrm{sim}}(5\,\mathrm{km})$ exceeds 30‰, meaning that the uncertainty of the retrieval simulations is of the same order as the uncertainty of the $\{H_2O,\delta D\}$ pairs remote sensing data product (Wiegele et al., 2014).

### 4.3   Interferences in retrieval simulations

As discussed in the context of Eqs. (2) and (4), the Jacobians and thus the averaging kernels depend on the satellite's viewing angle, surface conditions as well as atmospheric temperature and humidity profiles. In this section we examine to what extent

uncertainties in these parameters interfere with the averaging kernel simulations. Significant interferences would compromise the possibility of studying moisture pathway by $\{H_2O,\delta D\}$ pair inter-comparisons.

     We assume uncertainties in the parameters that describe the surface and atmospheric conditions according to the second column of Table 1. We calculate the interferences as the difference between the $s_{\mathrm{sim}}(5\,\mathrm{km})$ values obtained when changing the




different parameters by their uncertainty values. Table 1 lists the percentiles (the 50%-percentile means that for 50% of all the situations the respective interference is smaller than the value given in the table).

We observe that the interferences due to uncertainties in surface conditions (emissivity and temperature) are generally smaller than 3‰ (even the 95%-percentiles are smaller than 3‰). This means that even an error of 10% or 5 K in the surface emissivity

or temperature, respectively, do not significantly affect the simulation error. This is an important finding and means that our model-measurement comparisons will only be very weakly affected by uncertain model surface conditions. We observe the strongest interference due to uncertainty in the free tropospheric humidity levels. This is good, because it is the parameter that is most strongly linked with the $\{H_2O, \delta D\}$ pairs in the free troposphere.

We also estimate the interference due to using a constant satellite viewing angle of $25°$ instead of a variable angle. It is

very small (the 95%-percentile is at 1.3‰), meaning that we can use the constant satellite viewing angle of $25°$ without compromising the validity of our averaging kernel simulations.

## 5   Applications of the retrieval simulator

In this section we give some application examples of the retrieval simulator for validating the $\{H_2O, \delta D\}$ pair distributions obtained from the model ECHAM5-wiso (Werner et al., 2011). Here we work with model simulations that are nudged to large-

scale meteorological fields from ECMWF ERA-Interim (Butzin et al., 2014). This is not ideal for investigating how the model physics couple moisture pathways and large-scale atmospheric circulation, because nudging means that the large-scale temperatures and wind fields are no exclusive consequence of the modeled radiative and latent heat transfer, which in their turn are strongly connected to moisture transport pathways. Nevertheless, the nudged model run allows for an adequate demonstration of the working principle of the retrieval simulator, even by looking on a few days only. Such principle demonstration is the

objective of this section.

The following exemplary study will identify differences between the model and the observations thereby revealing the potential of the $\{H_2O, \delta D\}$ pair distributions. However, a more detailed analyses should be made in a dedicated study. Here we will only provide a very brief discussion of these differences.

Since the MUSICA MetOp/IASI $\{H_2O, \delta D\}$ product is only available for clear sky we need a clear sky criterion for the model

data. We define as a clear sky model situation the situations where the relative humidity in the model profile is smaller than 90% for all altitudes between the surface and 12 km a.s.l. For the retrieval simulations we use the vertical profiles of pressure, temperature and humidity mixing ratios as provided by ECHAM5-wiso, surface skin temperature from ECMWF ERA-Interim (http://apps.ecmwf.int/datasets/), emissivity data as used for our MUSICA MetOp/IASI retrievals (details see Wiegele et al., 2014) and a satellite observing angle fixed to $25°$.

### 5.1   Latitudinal and seasonal scale signals over the central Pacific

For our first exemplary study we investigate the $\{H_2O, \delta D\}$ pair distributions over the ocean for different latitudinal regions and seasons. We chose the central Pacific Ocean ($140°E$-$140°W$), since this is the region where effects from land/continent should





be very weak. We look on five different latitudinal belts and analyse the situation for mid February 2014 (12 Feb. to 18 Feb.) and mid August 2014 (12 Aug. to 18 Aug.). For this study we only work with IASI morning overpasses and with model data that correspond to local time between 6:00 and 12:00. Details about the calculation of the local times are given at the beginning of the next subsection.

Figure 11 depicts the contours indicating the highest density of the $\{H_2O, \delta D\}$ pairs. The contour lines mark the area that contains 66% of all $\{H_2O, \delta D\}$ pairs. The different colours of the contour lines represent the $\{H_2O, \delta D\}$ pairs density distribution for the different latitudinal belts as given in the legend. We use a logarithmic scale for the $H_2O$ concentrations, so that Rayleigh lines (lines that describe a processes where water is immediately removed when reaching the condensation point) appear as straight lines. The black dashed lines represent Rayleigh lines corresponding to the different initialisation conditions

and atmospheres that are representative for a very warm, temperately warm, and cold Ocean (corresponding water and boundary layer temperatures of 30, 20 and $10°C$). The left and right panels show the situations for mid February and mid August, respectively.

The top row of panels show the ECHAM5-wiso data for a broad layer around 5 km a.s.l.. We calculate the signals for this broad layer by convolving the ECHAM5-wiso profiles with a normalised Gauss function with the peak at 5 km a.s.l. and a

FWHM (full width half maximum) value of 5 km. These contour lines are calculated by considering all situations where the clear sky criterion is fulfilled.

The middle row of panels shows the ECHAM5-wiso data for 5 km a.s.l. after passing through the retrieval simulator, i.e. we applied Eq. (1) to the modeled states, whereby we use $\mathbf{A}$ as provided by the retrieval simulator. The respective contour lines are calculated for situations where the clear sky and sensitivity ($s_{err} < 50‰$) criteria are fulfilled. Especially for middle and

higher altitudes, the sensitivity criteria removes a lot of $\{H_2O, \delta D\}$ data pairs. This can be seen by comparing the numbers $N$ as given in the legends of the plots ($N$ gives the number of $\{H_2O, \delta D\}$ data pairs used for calculating the respective contours). The sensitivity filter removes mainly data that represent dry conditions and the distribution plots as shown in the middle panels only contain a significant number of $\{H_2O, \delta D\}$ data pairs if the $H_2O$ concentrations are larger than $10^3$ ppmv. The sensitivity filter explains most of the difference between the plots in the top and the middle row of the panels.

The bottom row of panels shows the contour plots as obtained from the MUSICA MetOp/IASI retrievals. These data are for clear sky situations (determined by EUMETSAT cloud filter and MUSICA MetOp/IASI retrieval fit quality filter) and fulfill the sensitivity criteria (we only work with data where we estimate a $s_{err} < 50‰$ from the actual averaging kernel). These data have the same characteristics as the ECHAM5-wiso driven retrieval simulations and we can compare the plots from the bottom row of panels with the plots from the middle row of panels.

We observe that the humidity concentrations are well predicted by the model. For the different seasons and latitudes the model and observation cover very similar humidity ranges. There is no significant difference between the modeled humidity and the humidity as observed by IASI. There is also some agreement concerning the latitudinal and seasonal variation of the $\{H_2O, \delta D\}$ pairs. For instance, in model and observation the tropical latitudinal belt ($10°S$-$10°N$, red contours) contains $\{H_2O, \delta D\}$ pairs with less negative $\delta D$ than the poleward latitudinal belts of the summertime hemisphere ($20°S$-$40°S$ in Febru-

ary, green contours, and $20°N$-$40°N$ in August, cyan contours). Another example are the wintertime hemispheres, where the



H$_2$O is relatively low but $\delta$D relatively high in the model as well as in the observation (the contours move to the upper left corner in the plots).

However, for humid situations (for H$_2$O concentrations above $10^{3.3}$ppmv $\approx$ 2000ppmv) the modeled $\delta$D values are significantly less negative then the measured $\delta$D values. In the model data increased humidity means also less negative $\delta$D values, i.e. the {H$_2$O,$\delta$D} pairs belonging to a certain latitudinal belt are well distributed along a Rayleigh line. This is different in the measured: data belonging to the same latitudinal belt have almost the same $\delta$D values for low and high humidity. We observe that the slopes of the {H$_2$O,$\delta$D} pair distributions of the measured data are rather shallow and cross different Rayleigh lines, whereas the model predicts slopes that are reasonably in parallel to a single Rayleigh line.

## 5.2 Diurnal scale signals over the Tropics and Subtropics

In the second exemplary study we investigate diurnal scale signals in the {H$_2$O,$\delta$D} pair distribution. Such signals are seen in the MUSICA MetOp/IASI data (e.g. bottom panel of Fig. 10 of Schneider et al., 2016) and here we analyse morning and evening signals obtained from the IASI observations together with respective signals in the ECHAM5-wiso model data. IASI has a morning overpass at about 9:30 local time and a evening overpass at about 21:30 local time. ECHAM5-wiso provides global outputs for 6:00, 12:00, 18:00 and 24:00 universal time, and we calculate the local time by adding $\mathrm{lon}/15°$ to the universal times (whereby $\mathrm{lon}$ are longitudes in degree, with eastern longitudes from 0° to 180° and western longitudes from 0° to $-180°$).

The study is made for the tropics (February and August, latitudinal belt from 10°S to 10°N), whereby we distinguish between data that represent the atmosphere over ocean and over land (see left panels in Fig. 12), and for the subtropics in the summertime hemisphere (mid August, latitudinal belt from 22°S to 35°N), whereby we separately consider the central Atlantic and the Sahara desert (see right panels in Fig. 12). The order from the top to the bottom is as in the previous figure: ECHAM5-wiso data for a broad layer around 5 km a.s.l., ECHAM5-wiso driven retrieval simulations and MUSICA MetOp/IASI retrieval results. The blueish coloured contour lines represent the ocean scenarios and the greenish coloured contour lines the land scenarios. The bright colours represent morning (mor) and the dark colours for evening (eve) data.

The comparison of the first two rows of panels gives insight into the effect of the retrieval simulator. There are only a few moderately dry situations that are removed by the sensitivity filter ($s_{\mathrm{err}} < 50$‰) and consequently the effect of the retrieval simulator is significantly smaller in these regions than for the higher latitudes analysed in the context of Fig. 11. It seems that in the tropics and summertime subtropics the model generates atmospheric states for which IASI can reasonably well observe free tropospheric {H$_2$O,$\delta$D} pairs.

The plots in the middle and bottom row of panels are for data that have the same characteristics and they can be compared in a meaningful way. There are several similarities to the comparison made in the context of the previous figure: (a) again the slopes of the {H$_2$O,$\delta$D} pair distribution of the model are in parallel to a Rayleigh line, whereas in the observational data the slopes are generally rather shallow, (b) again we find that the model and the IASI retrieval produce relatively similar humidity concentrations (in the tropics and in the summertime subtropics over the Atlantic).



However, the subtropical summertime atmosphere of the Sahara desert is significantly moister in the model than in the IASI data (compare the greenish contours in the right panels). Interestingly, the Sahara is also the region where IASI observes a very strong diurnal signal. While IASI observes rather similar humidity concentrations for morning and evening, the $\delta$D values are significantly less negative in the evening than in the morning. Such HDO enriched air over the Sahara has been reported in a variety of studies (Schneider et al., 2015; Dyroff et al., 2015; González et al., 2016; Schneider et al., 2016) and can be attributed to upward mixing of boundary layer air with dry free tropospheric air driven by the Saharan heat lows. It seems that this mixing develops during the day (the strongest effects are seen in the evening), whereas during the night the subtropical subsidence circulation dominates bringing the atmosphere at 5 km a.s.l. back to the free tropospheric subtropical background (which is then observed in the morning). This pronounced diurnal signal over the Sahara is not seen in the ECHAM5-wiso data. Concerning the subtropical latitudinal belt over the central Atlantic (see blueish contours) IASI and ECHAM5-wiso consistently observe no diurnal signal.

The left panels show that for tropical ocean scenarios model and satellite consistently find no diurnal signal. However, there seems to be a weak diurnal cycle over land, where the MUSICA MetOp/IASI morning data show slightly higher humidity concentrations than the evening data with at the same time almost unchanged $\delta$D values. We think that this can be explained by the IASI cloud filter: convection over the tropical land increases the cloudiness during the day. These clouds rain out during the late afternoon and night and cloudiness decreases again until the morning hours. Then the satellite can observe those regions where it was raining during the night. At these locations the air is humid and the relatively negative $\delta$D for high humidity concentration might indicate that evaporation of falling rain is an important source of this moisture (please recall Fig. 1, showing that the values retrieved at 5 km a.s.l. have also significant contributions from 2 km a.s.l. and below). The respective diurnal signal is not seen in the ECHAM5-wiso data. In this context it is also interesting that the model has almost the same number of acceptable data (data that pass the cloud and sensitivity filter) in the evening and morning. This is in clear contrast to the IASI data. There are much more acceptable data in the morning than in the evening, because the IASI cloud filter removes much more data for the evening overpass than for the morning overpass. It seems that the missing diurnal signal in the modeled $\{H_2O,\delta D\}$ pair distribution is somehow related to the missing diurnal cycle in the cloudiness. A reason might be a poor timing of the cloud formation in the model.

## 5.3 Brief discussion of the major model-measurement differences

The two exemplary comparisons in Figs. 11 and 12 show significant differences in the slopes of the $\{H_2O,\delta D\}$ pair distribution. Concerning the measurements, data belonging to a certain latitudinal belt and season are distributed over different Rayleigh lines, indicating that low and high humidity can be explained by cold and warm ocean sources or that evaporation of falling rain plays an important role for moistening the free troposphere. In the model the $\{H_2O,\delta D\}$ pairs that belong to a certain latitudinal belt are distributed along a single Rayleigh line. It seems as if in the model all the water mass observed for a certain latitudinal belt has a very similar source region. However, the $\{H_2O,\delta D\}$ pairs of the tropical belt are distributed along a Rayleigh line that corresponds to a ocean temperature of only 20°C. It is the same Rayleigh line as for the mid-/high-latitudinal belts (65°-45°) of the summer time hemisphere.



It is unlikely that tropical water masses have no ocean source corresponding to temperatures above 20-25°C and that the source region is almost the same as for a mid- and high-latitudinal water mass. So it is unlikely that the modeled $\{H_2O,\delta D\}$ pair distributions are solely explained by a Rayleigh process. Another important process for free tropospheric humidity might be vertical mixing. Actually Risi et al. (2012) found for the LMDZ-iso model that excessive vertical transport in the model

can explain too strong HDO enhancement in the model, suggesting that excessive vertical transport in ECHAM5-wiso is also responsible for the difference between ECHAM5-wiso and IASI for high humidity. Furthermore, an excessive vertical transport might also explain why the model fails in capturing the diurnal signal over the Sahara. This signal is due to vertical mixing that develops during the day. Since this mixing is always more or less existent in the model, it is not able to capture the additional diurnal time scale mixing. The fact that the IASI evening observation, where mixing occurs, well agrees with the

model simulations (compare right panels in the second and bottom row of Fig. 12) clearly supporting this interpretation.

Our interpretation of the weak diurnal cycle as seen in the IASI data over tropical land (bottom left panel of Fig. 12) being connected to convection, cloud formation and rain, suggests that mid-tropospheric $\{H_2O,\delta D\}$ pair data can help improving the parameterisation of convection. This is consistent to the findings of Field et al. (2014) for GISS ModelE. Again, a too strong convective mixing in the ECHAM5-wiso model strongly dampens the occurrence of such diurnal cycle in the simulation.

**6  Summary and outlook**

The MUSICA MetOp/IASI retrieval scheme can generate reliable free tropospheric $\{H_2O,\delta D\}$ pairs for each morning and evening, for many years and on quasi global scale. Since the $\{H_2O,\delta D\}$ pairs record the water mass history, these MUSICA data can help investigating the links between moisture pathways and atmospheric circulation on different scales, thereby addressing a major challenge of climate research.

However, remote sensing $\{H_2O,\delta D\}$ pairs are complex data products. In the meantime these complexities are well-understood, but in order to be able to use the data in the context of comprehensive model studies, we need model data that have these complexities assimilated. Here we provide a tool that is needed for this assimilation step. It is a retrieval simulator that predicts the remote sensing averaging kernels that would result for a retrieval made for the model state (atmosphere and surface states). The tool is made available as a MATLAB and Python routine and can easily be connected to any model and the predicted kernels

can then be used for generating model data with the characteristics of the remote sensing data.

The retrieval simulator is based on the physical principles of atmospheric radiative transfer. It is shown that a consideration of these physical principles is necessary in order to understand the remote sensing measurement and thus in order to be able to simulate the averaging kernels.

The quality of the retrieval simulations is empirically assessed. It is shown that the simulator well identifies the situations in

which the remote sensing system is sensitive with respect to $\{H_2O,\delta D\}$ pairs and in which it is not. Furthermore, we are able to demonstrate that the simulator is even able to predict the altitude region where the remote sensing system is most sensitive for given atmospheric and surface conditions. In summary we can reliably predict the complexities of the remote sensing data for any atmospheric and surface situation around the globe. The uncertainties of the simulations are generally smaller than the





uncertainty of the remote sensing data product (i.e. in this sense they are not significant). Only for evening observations over land the uncertainty of the simulations can be of the order of the uncertainty of the remote sensing data.

We give a few example of the working principle of the simulator and apply it to ECHAM5-wiso data. We document that model data that have been processed with the simulator can be compared to the MUSICA MetOp/IASI {$H_2O$,$\delta D$} pair data. A

detailed analyses of these comparisons is out of the scope of this paper, but already the few examples suggest large potential of the method for evaluating the moisture pathways in atmospheric models.

Now that more and more atmospheric models have the isotopologues included, it is time to think in detail about the kind of atmospheric moisture processes that can be investigated with such models. For such investigations the model data need to be combined with reliable reference measurements. Our retrieval simulator allows such combination and can be easily adopted

to any model data. The characteristics of the MUSICA MetOp/IASI {$H_2O$,$\delta D$} pair product can be assimilated into model data independently from the availability of MUSICA MetOp/IASI data. In this sense the modelers can test different model setups and investigate to what extent the resulting {$H_2O$,$\delta D$} pair signal will be detectable by MetOp/IASI. Then they can give feedback to the remote sensing scientists who can work on dedicated retrievals that allow an evaluation of the modeled scenarios. Our vision is that the possibility of performing model tests and at the same time being able to check if there is a

chance to observe the modeled scenarios by IASI does strongly stimulate research in this field.

*Author contributions.*  C. Borger, A. Wiegele, O. E. García and E. Sepúlveda worked on the MUSICA MetOp/IASI retrievals and contributed to the analyses of the retrieval products. F. Hase developed the PROFFIT-nadir retrieval code. M. Werner developed the ECHAM water isotopologue products and provided ECHAM5-wiso data. M. Schneider coordinated and designed the MUSICA project, developed the retrieval simulator and prepared the manuscript with contributions from all co-authors.

*Acknowledgements.*  This study has been conducted in the framework of the project MUSICA which is funded by the European Research Council under the European Community's Seventh Framework Programme (FP7/2007-2013) / ERC Grant agreement number 256961.

E. Sepúlveda is supported by EUMETSAT (Fellowship Programme, project VALIASI).

We acknowledge the support by the Deutsche Forschungsgemeinschaft and the Open Access Publishing Fund of the Karlsruhe Institute of Technology.

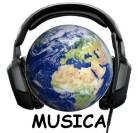
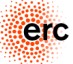
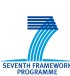
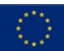



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





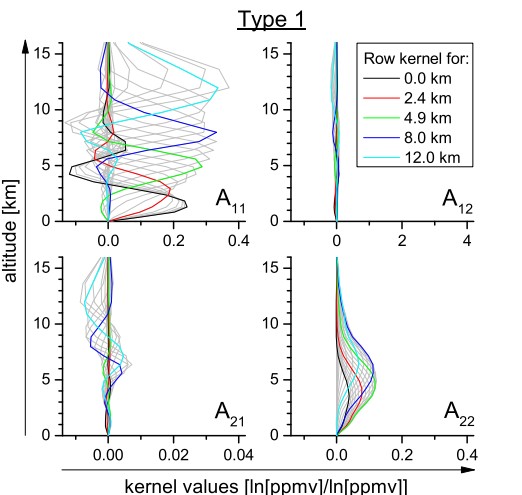

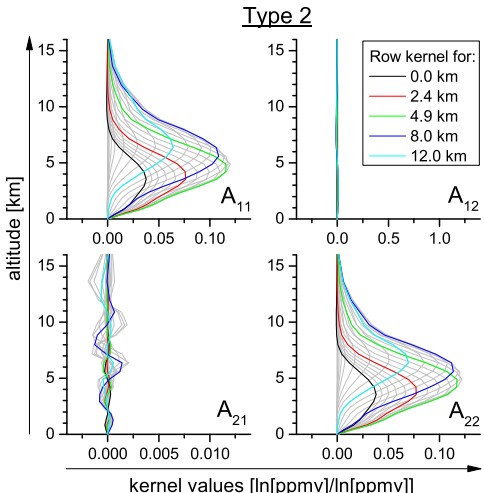

**Figure 1.** Example of an averaging kernel for an observation over the tropical ocean. Shown are the row entries of the four blocks of the MetOp/IASI full averaging kernel matrix in the {$H_2O$,$\delta D$} proxy basis system. Left for the Type 1 product (original retrieval output) and right for the Type 2 product (after aposteriori correction).




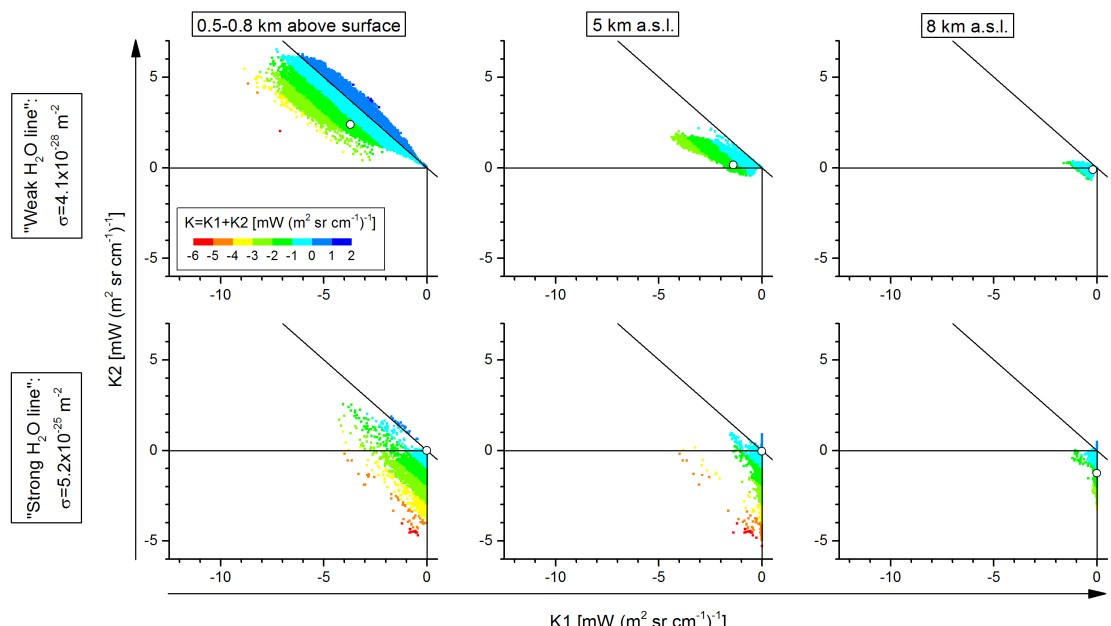

**Figure 2.** $H_2O$ Jacobians calculated according to Eqs. (2)-(4) for two different line strength and three different altitudes. Plotted is $K1_{i,j}$ versus $K2_{i,j}$ (the colour code informs about $K_{i,j} = K1_{i,j} + K2_{i,j}$). The upper panels represent a weak and a strong line, respectively. The left panels are for the boundary layer (about 500 to 800 m above surface), the middle panels for 5 km a.s.l., and the right panels for 8 km a.s.l.. The black circles filled by white colour indicate the results obtained for the surface and atmospheric conditions corresponding to the averaging kernels of Fig. 1.



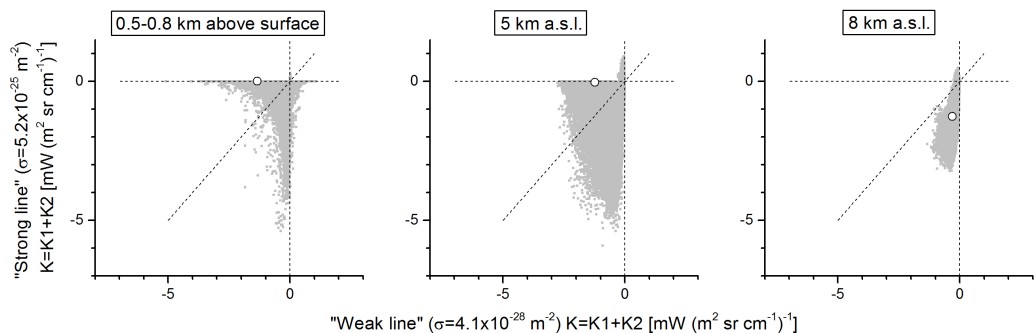

**Figure 3.** Relation between the Jacobians ($K_{i,j} = K1_{i,j} + K2_{i,j}$) obtained for the two different line strength and the three different altitudes from Fig. 2. The left panel is for the boundary layer (about 500 to 800 m above surface), the middle panel for 5 km a.s.l., and the right panel for 8 km a.s.l.; The black circles filled by white colour indicate the results obtained for the surface and atmospheric conditions corresponding to the averaging kernels of Fig. 1.





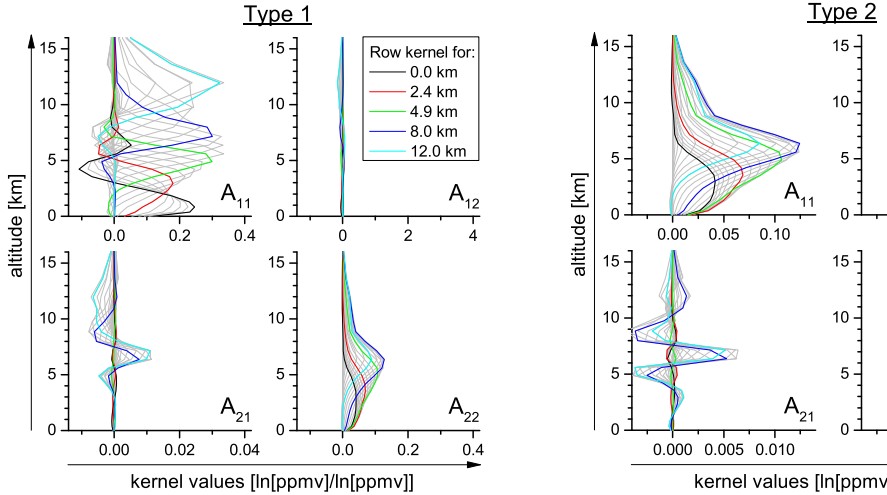

**Figure 4.** Simulated averaging kernels for the surface and atmospheric conditions corresponding to the averaging kernels of Fig. 1. Panels and colour are as in Fig. 1.





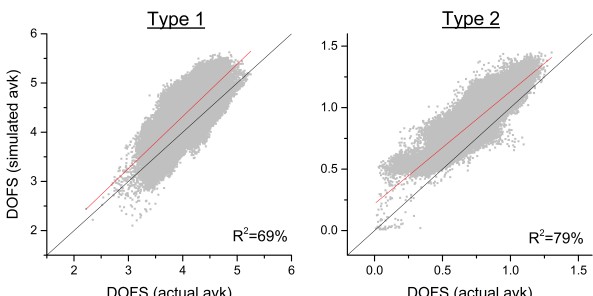

**Figure 5.** Comparison of the DOFS values obtained from the actual and the simulated averaging kernels. Left for the Type 1 product and right for the Type 2 product.





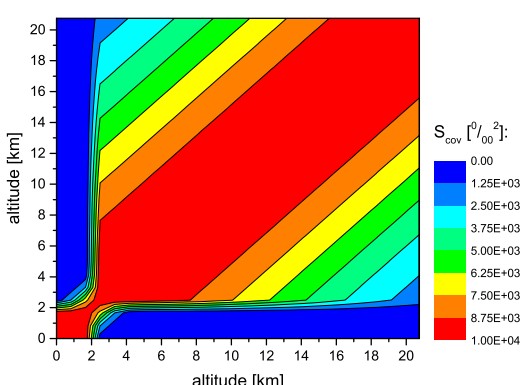

**Figure 6.** Covariances used for estimating the quality of the simulated averaging kernels.





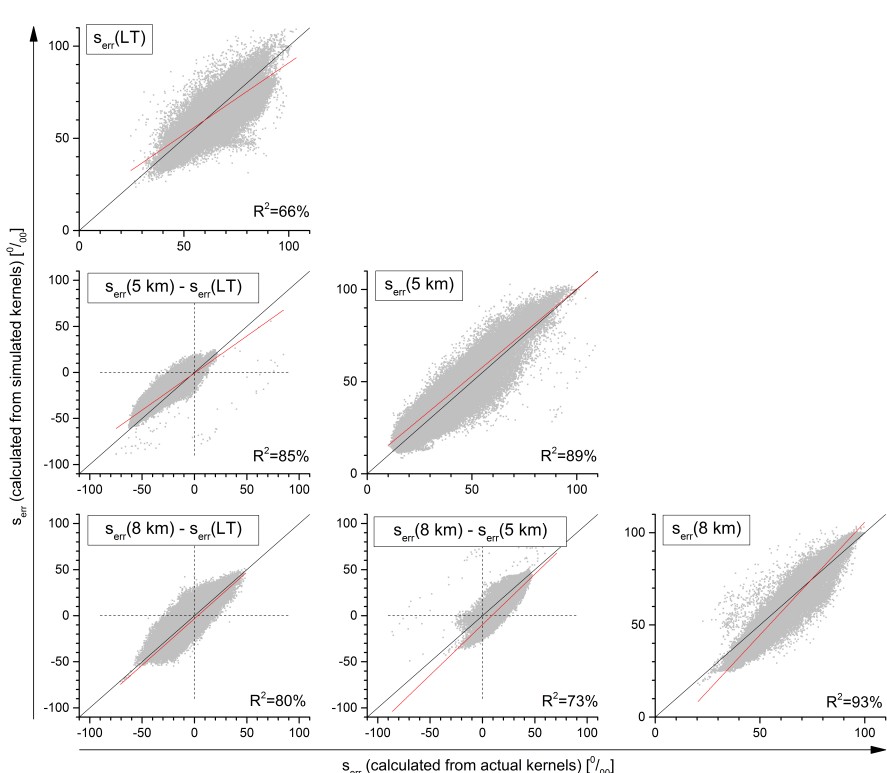

**Figure 7.** Evaluation of the simulated sensitivity error matrix $\mathbf{S_{err}}$ (calculation of $\mathbf{S_{err}}$ according to Eq. 7). Compared are of the square roots of the diagonal elements ($s_{err}$) as obtained from the actual and the simulated kernels. The legend informs about the six different $s_{err}$ products that are compared in the six different panels.





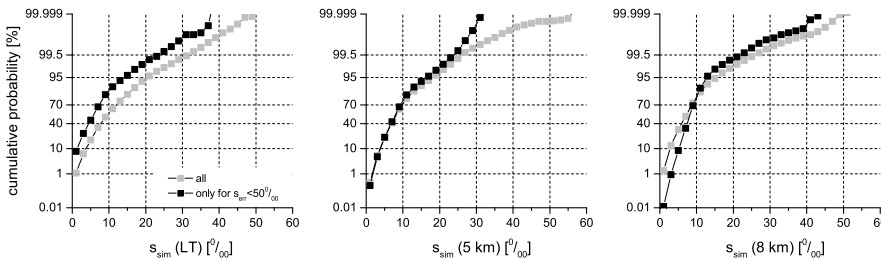

**Figure 8.** Cumulative occurrences of the $s_{sim}$ value obtained for 1500-2000 m above surface, 5 km a.s.l. and 8 km a.s.l. (from the left to the right, respectively). The calculations of $\mathbf{S_{sim}}$ are made according to Eq. 8. The grey dots are for all data and the black dots only for situations with good sensitivity $s_{err} \leq 50‰$).





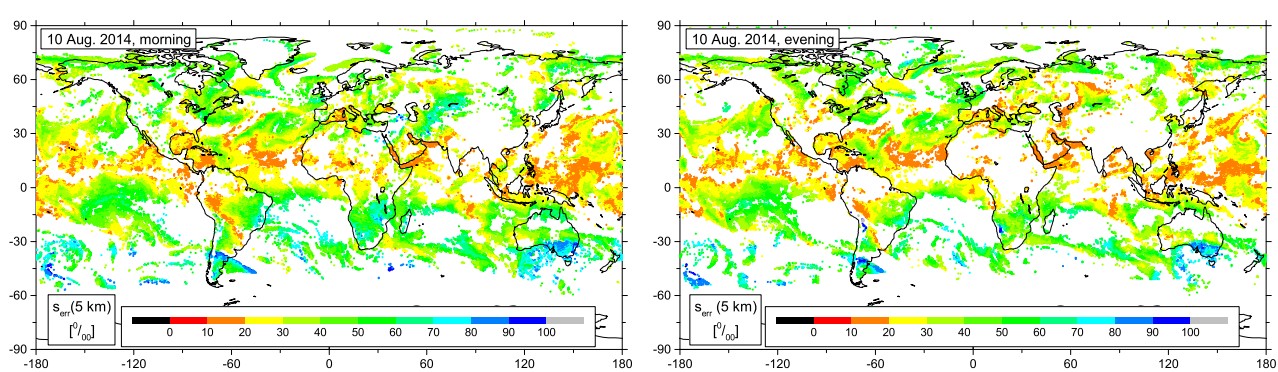

**Figure 9.** Example of the daily geographical distribution of $s_{\mathrm{err}}(5\,\mathrm{km})$ (calculation of $\mathbf{S_{err}}$ according to Eq. 7). Left panels show the morning overpass and the right panels the evening overpass for 10 Aug. 2014.





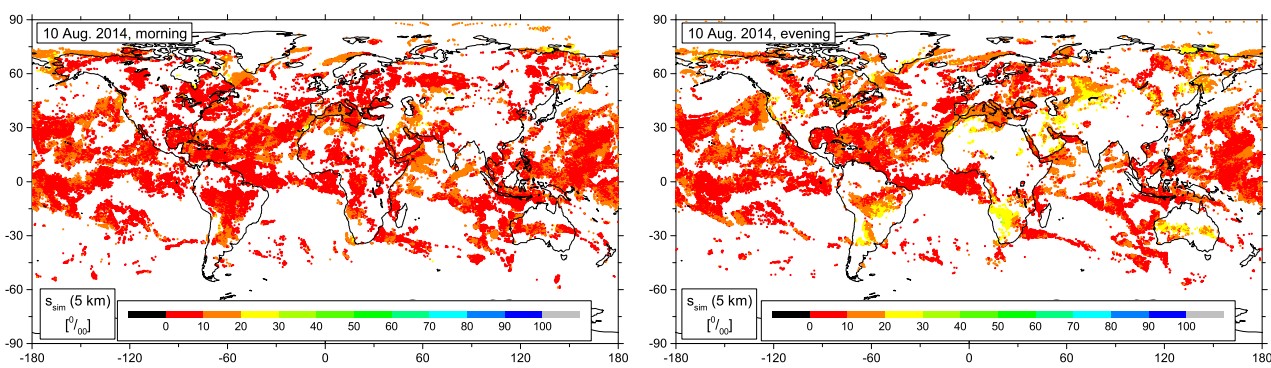

**Figure 10.** Example of the daily geographical distribution of $s_{sim}(5\,km)$ for those situations that fulfill the sensitivity criterion ($s_{err}(5\,km) <$ 50‰). The calculation of $\mathbf{S_{sim}}$ is made according to Eq. 8. Left panels show the morning overpass and the right panels the evening overpass for 10 Aug. 2014, i.e. same coverage as in Fig. 9.





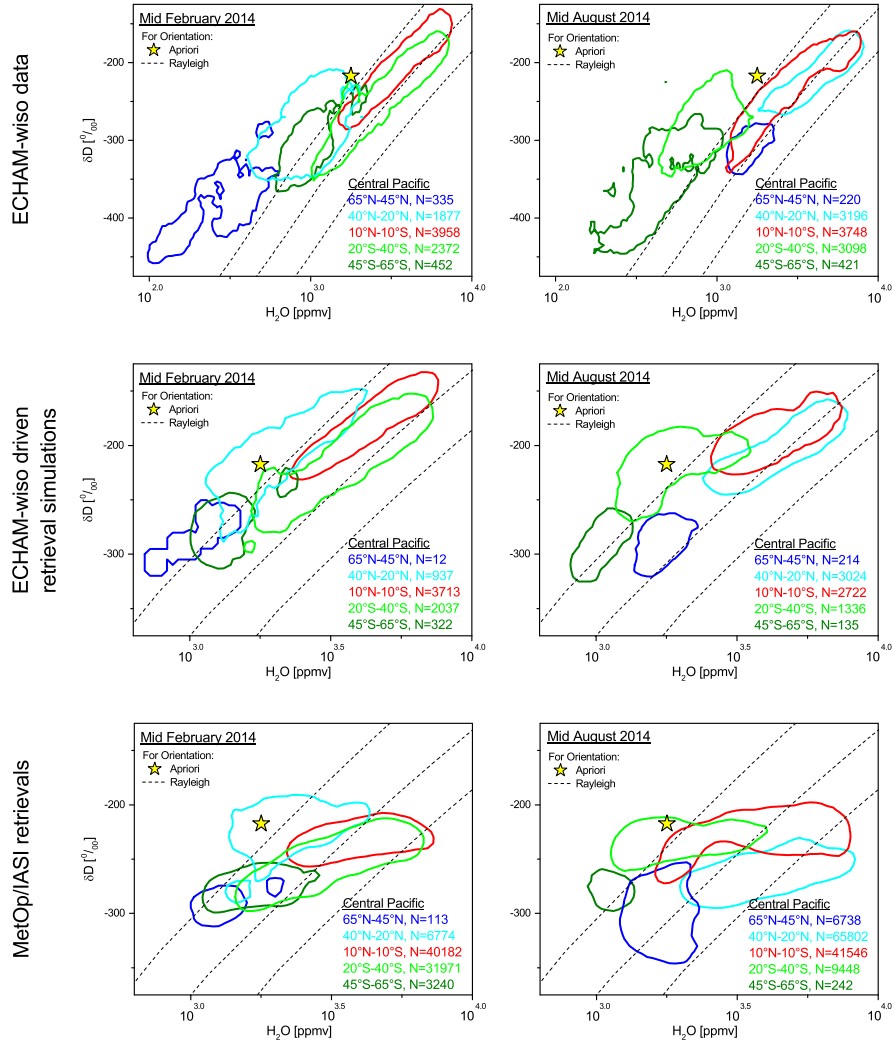

**Figure 11.** Latitudinal and seasonal scale signals in the {$H_2O$,$\delta D$} pair distribution in the free troposphere over the central Pacific (140°E-140°W). From the top to the bottom: distributions obtained from ECHAM-wiso simulations, from ECHAM-wiso simulations after passing through the retrieval simulator and from the MetOp/IASI retrievals, respectively. Left panels for mid February and right panels for mid August. The different colours represent the different latitudinal belts as given in the legend. The black dashed lines represent Rayleigh curves for three different initialisations: {T = 10°C; RH = 60%; $\delta D = -89‰$}, {T = 20°C; RH = 80%; $\delta D = -78‰$} and {T = 30°C; RH = 100%; $\delta D = -69‰$}. The yellow star marks the apriori value used for the remote sensing retrievals at 5 km. Please note the different scale for the plots in the top panels.





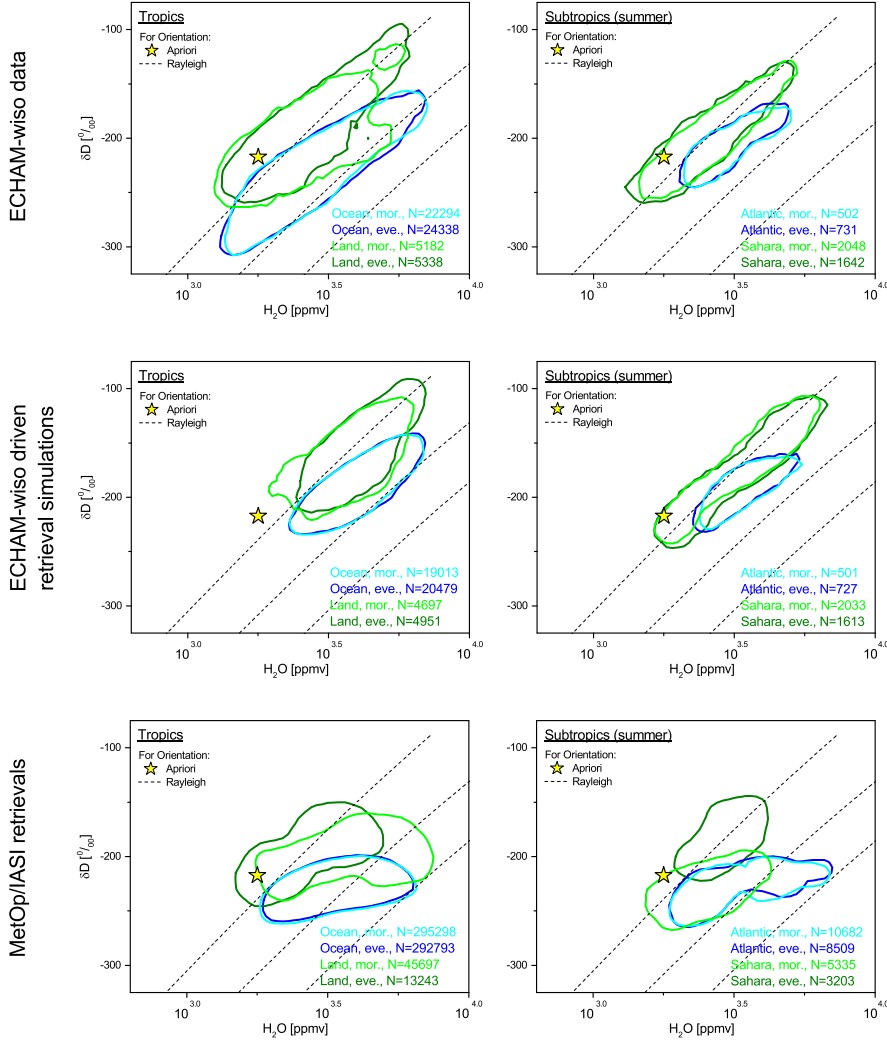

**Figure 12.** Diurnal scale signals in the $\{H_2O, \delta D\}$ pair distribution for different scenarios. From the top to the bottom as in Fig. 11: distributions obtained from ECHAM-wiso simulations, from ECHAM-wiso simulations after passing through the retrieval simulator and from the MetOp/IASI retrievals, respectively. Left panels for land and ocean scenarios in the Tropics (Feb.+Aug., $10°$S-$10°$N) and right panels for the summertime Subtropics (Aug., $22.5°$N-$35°$N) over the central Atlantic ($30°$W-$50°$W) and the Sahara/Arabia ($10°$W-$50°$E). The different colours represent the different day times (morning, $6\,$LT to $12\,$LT, and evening, $18\,$LT to $24\,$LT) and scenarios as as given in the legend. The black dashed lines represent the same Rayleigh curves as in Fig. 11.



**Table 1.** Interferences in $s_{\mathrm{sim}}(5\,\mathrm{km})$ due to uncertainties in surface and atmospheric conditions. Listed are the 50%-, 80%- and 95%-percentiles (P50, P80 and P95, respectively) for land and ocean scenarios.

| Source | Assumed uncertainty | Interference over land P50 / P80 / P95 | Interference over ocean P50 / P80 / P95 |
|---|---|---|---|
| Surface emissivity | 10% | 1.1‰ / 1.8‰ / 2.5‰ | 0.8‰ / 1.4‰ / 2.3‰ |
| Surface temperature | 5 K | 1.2‰ / 2.0‰ / 2.9‰ | 0.8‰ / 1.5‰ / 2.6‰ |
| Boundary layer temperature | 5 K | 1.4‰ / 2.0‰ / 2.5‰ | 1.8‰ / 2.4‰ / 2.7‰ |
| Free tropospheric temperature | 5 K | 1.4‰ / 2.5‰ / 3.2‰ | 1.6‰ / 2.3‰ / 3.1‰ |
| Upper tropospheric temperature | 5 K | 0.6‰ / 1.2‰ / 1.7‰ | 0.5‰ / 1.2‰ / 2.0‰ |
| Boundary layer humidity | 25% | 2.4‰ / 3.2‰ / 3.7‰ | 2.8‰ / 3.2‰ / 3.5‰ |
| Free tropospheric humidity | 25% | 2.0‰ / 3.0‰ / 4.0‰ | 2.2‰ / 3.2‰ / 4.0‰ |
| Upper tropospheric humidity | 25% | 0.9‰ / 1.7‰ / 2.0‰ | 1.7‰ / 2.1‰ / 2.4‰ |