# Peer review of "MUSICA MetOp/IASI $\{H_2O, \delta D\}$ pair retrieval simulations for validating tropospheric moisture pathways in atmospheric models"

_Atmospheric Measurement Techniques, 2016_

## Referee Comment (RC1) · Anonymous Referee #1 · 18 Nov 2016

Schneider et al have developed a forward-model based approach to predicting the averaging kernels from a model's state, required in applying instrument operators for comparing modeled H20-dD pairs to those from the MUSICA retrievals. This is a novel approach and is a nice advance beyond previous statistical-based approaches. The paper provides detailed diagnosis of the retrieval simulator performance, and I have no major technical concerns. I recommend publication as-is, subject to the following minor issues.

P2L26: suggest saying 'nudged to meteorology or forced by prescribed ocean temperatures'

P3L16: throughout, please replace 'exemplary' with something else such as "case-

study" or "example"

P6L7: here and elsewhere, suggest replacing 'transport' with 'transfer'

P9L9: for example, replace "exemplary tropical ocean situation" with "tropical ocean case"

P9L10: replace "(compare Figs. 1 and 4)" with "(seen by comparing Figs. 1 & 2)

P9L30: throughout, replace "well capture" or similar phrases with, in this case, "Our simulations capture the actual sensitivity of the remote sensing system reasonably well"

P11L3: Insert 'The' before 'Best sensitivity'

P11L29: Omit "interesting"

P13L19: Can omit the "Such principle demonstration" sentence.

P14L5: In this section, please provide a few referenced sentences on interpretation of H20-dD pairs with Rayleigh curves

P15L6: omit ':' in 'measured: data'

P17L20: Suggest omitting the phrase "In the meantime these complexities are well-understood, but' and begin the sentence with 'In order to...'

P17L29: similar to 'well capture', please rephrase to omit 'well identifies'

P18L3: change 'example' to 'examples'

P18: In section 6 (or even in the introduction), please consider a paragraph situating your approach in the context of other retrieval simulator efforts, namely for cloud simulators in general (e.g. Bodas-Salcedo, 2011, BAMS for COSP) and in the potential for your approach to be applied for other trace gases, extending, for example the statistical approach of Worden et al. (2013, AMT) for CO and O3. Your approach for H2O-dD pairs I think is relevant to the latter and it is worth making that point.

P27: Suggest replacing second sentence in figure caption with 'The legends indicate the six...'

---

## Referee Comment (RC2) · Anonymous Referee #2 · 19 Dec 2016

In this manuscript the authors present a new technique for mapping simulated water vapor isotopic composition from GCM output into simulated remote sensing data for the MetOp/IASI platform. It's a very important technique and it represents an advance over previous methods developed for other remote sensing platforms.

My specialty is in the applications of this kind of data, and not in the remote sensing techniques that make up the core of the paper. I hope the editors are able to secure a review from someone who can evaluate those more technical aspects of the paper. The 'Applications' part of the manuscript is simply attempting to demonstrate that the authors' technique works and is not trying to address any scientific question, so that component is fine as far as it goes and I look forward to seeing more applications of

this technique in forthcoming papers.

My only real request is to include a more functional example for the Matlab code in the supplemental materials. I took a look at the Matlab code that was provided and found the f_IASIavkT2simulation_v160706.m file, which is a matlab function. That's fine, and the code is clear and well written. I think this contribution would benefit from another Matlab script that uses the function in an example. For those of us who are not specialists in remote sensing, such an example would go a long way to helping the authors' excellent work be available to a broader audience. If there's a good working example, it's easier for the community to modify it to their own applications.

Otherwise, I think this is a fine contribution to AMT. Once the authors are able to put together a good working example for the Matlab and Python code, the paper will be ready for publication.
* * *

---

## Author Comment (AC1) · 14 Jan 2017

Dear Referees,
Dear Editor,

Thank you for your efforts in reviewing and commenting our paper. Below we give answers to all your comments (in blue fonts).

Referee #1:
Schneider et al have developed a forward-model based approach to predicting the averaging kernels from a model's state, required in applying instrument operators for comparing modeled H20-dD pairs to those from the MUSICA retrievals. This is a novel approach and is a nice advance beyond previous statistical-based approaches. The paper provides detailed diagnosis of the retrieval simulator performance, and I have no major technical concerns. I recommend publication as-is, subject to the following minor issues.

P2L26: suggest saying 'nudged to meteorology or forced by prescribed ocean temperatures'
Ok.

P3L16: throughout, please replace 'exemplary' with something else such as "case-study" or "example"
Ok, we will avoid "exemplary" and replace it by your suggested terminology.

P6L7: here and elsewhere, suggest replacing 'transport' with 'transfer'
Yes, saying "radiative transfer" is better than saying "radiative transport". We will use consistently and throughout the paper the terminology "radiative transfer".

P9L9: for example, replace "exemplary tropical ocean situation" with "tropical ocean case"
Ok, we will avoid "exemplary" and think in alternatives as you suggest.

P9L10: replace "(compare Figs. 1 and 4)" with "(seen by comparing Figs. 1 & 2)
Ok, we will write "(seen by comparing Figs. 1 and 4)".

P9L30: throughout, replace "well capture" or similar phrases with, in this case, "Our simulations capture the actual sensitivity of the remote sensing system reasonably well"
Ok, we will relativize respective statements by using "reasonably well" instead of "well".

P11L3: Insert 'The' before 'Best sensitivity'
Ok.

P11L29: Omit "interesting"
Yes "interesting" can be omitted there because that the situations are the "interesting ones" is already stated two lines before.

P13L19: Can omit the "Such principle demonstration" sentence.
Honestly we would like to leave this sentence there or maybe change it slightly. What we want to express is that the potential of the method lies in working with large datasets and with free model runs. Here we work with a small data set and therefore need a nudged model run. This is sufficient to demonstrate the working principle of the method, because for large data sets and free model runs (the really interesting studies!) the working principle will be the same.

P14L5: In this section, please provide a few referenced sentences on interpretation of H20-dD pairs with Rayleigh curves
Ok, we will try to give a very brief introduction into {H2O,delD} pair distribution for a pure Rayleigh process and reference to pioneering work from the 1960s. Please also note, that a discussion/interpretation of the observed and modeled {H2O,delD} pair distribution by means of theoretical Rayleigh lines is provided in Section 5.3 (e.g. in the first two paragraphs of the section).

P15L6: omit ':' in 'measured: data'
It should be "[…] in the measurement: data belonging to […]".

P17L20: Suggest omitting the phrase "In the meantime these complexities are well understood, but' and begin the sentence with 'In order to. . .'

We would like to leave it, because we can only correctly assimilate the complexities of the {H2O,delD} pair remote sensing data to the model data if we really understand these complexities. This is not trivial and only very recently methods for capturing/fully describing the complexities of {H2O,delD} pair remote sensing data have been developed (see review of Schneider et al., 2016). So in our opinion this statement is important here.

P17L29: similar to 'well capture', please rephrase to omit 'well identifies'
Ok, we will also relativize and use the expression "reasonably well".

P18L3: change 'example' to 'examples'
Ok.

P18: In section 6 (or even in the introduction), please consider a paragraph situating your approach in the context of other retrieval simulator efforts, namely for cloud simulators in general (e.g. Bodas-Salcedo, 2011, BAMS for COSP) and in the potential for your approach to be applied for other trace gases, extending, for example the statistical approach of Worden et al. (2013, AMT) for CO and O3. Your approach for H2O-dD pairs I think is relevant to the latter and it is worth making that point.

We present a predictor for tropospheric humidity kernels obtained from well-resolved thermal nadir spectra of IASI. Our predictor simulates the radiance measurements and the retrieval process: we perform forward calculation (although very simplified) and consider the constrained inversion process. In the Introduction Section (page 3, lines 3-10) we situate our work in the context of the work by Field et al. (2012), which presents another predictor for a tropospheric humidity kernel (actually for HDO only) obtained by retrievals of the well resolved thermal nadir spectra of TES. They use a multi regression method to predict the kernels from a few atmospheric and surface state parameters, i.e. it is a statistically-based kernel predictor. In our opinion such predictors for humidity kernels are especially important, because of the large variations of tropospheric humidity (large amplitude of variations taking place on rather small spatial and temporal scales). We are not aware of any other predictor for tropospheric humidity kernels for retrievals that use well resolved thermal nadir spectra.

Worden et al. (2013) presents a kernel predictor for CO and O3. Their approach is also based on a multi-regression method, i.e. it is also a statistical method: the regression coefficients for different atmospheric and surface state parameters are obtained by requiring best representation of the kernels for a training data set. To our understanding, the main difference to Field et al. (2012) is the much higher number of parameters used for the multi-regression fit.
Bodas-Salcedo et al. (2011) present a very comprehensive software package (COSP: CFMIP Observation Simulator Package, whereby CFMIB stands for Cloud Feedback Model Intercomparison Project) that can simulate kernels for many different satellite instruments. The focus is on the products related to precipitation, cloud distribution and cloud properties. For some instruments/retrievals COSP simulates the measurement (forward calculations) and then the retrievals process, i.e. it is similar to our approach. For other instruments/retrievals COSP uses statistical methods that are similar to the Field et al. (2012) and Worden et al. (2013) approaches.
When preparing the revised manuscript we will have a more detailed look on the Bodas-Salcedo (2011) and Worden et al. (2013) papers and try to very briefly discuss/reference these works in one or two sentences. We will insert the respective discussion in the introduction section in the same paragraph where we discuss the Field et al. (2012) work.
.
P27: Suggest replacing second sentence in figure caption with 'The legends indicate the six. . .'
Ok.

Referee #2:

In this manuscript the authors present a new technique for mapping simulated water vapor isotopic composition from GCM output into simulated remote sensing data for the MetOp/IASI platform. It's a very important technique and it represents an advance over previous methods developed for other remote sensing platforms. My specialty is in the applications of this kind of data, and not in the remote sensing techniques that make up the core of the paper. I hope the editors are able to secure a review from someone who can evaluate those more technical aspects of the paper. The 'Applications' part of the manuscript is simply attempting to demonstrate that the authors' technique works and is not trying to address any scientific question, so that component is fine as far as it goes and I look forward to seeing more applications of this technique in forthcoming papers. My only real request is to include a more functional example for the Matlab code in the supplemental materials. I took a look at the Matlab code that was provided and found the f_IASIavkT2simulation_v160706.m file, which is a matlab function. That's fine, and the code is clear and well written. I think this contribution would benefit from another Matlab script that uses the function in an example. For those of us who are not specialists in remote sensing, such an example would go a long way to helping the authors' excellent work be available to a broader audience. If there's a good working example, it's easier for the community to modify it to their own applications. Otherwise, I think this is a fine contribution to AMT. Once the authors are able to put together a good working example for the Matlab and Python code, the paper will be ready for publication.

We are very happy with a comment coming from a potential user of the retrieval simulator and we are very interested in making the simulator as easy to use as possible. So following your suggestion we will provide additional supplements:

1. a netcdf-file with example model data.
2. a "working example" code (Matlab and Python) that reads the model data from the netcdf-file and outputs the model data after having passed through the retrieval simulator.